# Anthropogenic Stressors in Upland Rivers: Aquatic Macrophyte Responses. A Case Study from Bulgaria

**DOI:** 10.3390/plants10122708

**Published:** 2021-12-09

**Authors:** Gana Gecheva, Karin Pall, Milcho Todorov, Ivan Traykov, Nikolina Gribacheva, Silviya Stankova, Sebastian Birk

**Affiliations:** 1Faculty of Biology, Plovdiv University, 4000 Plovdiv, Bulgaria; n.gribacheva@mail.bg (N.G.); stankova_1991@abv.bg (S.S.); 2Systema GmbH, 8 Bensasteig, 1140 Vienna, Austria; karin.pall@systema.at; 3Institute of Biodiversity and Ecosystem Research, BAS, 1113 Sofia, Bulgaria; todorovmilcho@gmail.com; 4Faculty of Biology, Sofia University, 1164 Sofia, Bulgaria; itraykov@biofac.uni-sofia.bg; 5Faculty of Biology, University of Duisburg-Essen, 45141 Essen, Germany; sebastian.birk@uni-due.de; 6Centre for Water and Environmental Research, University of Duisburg-Essen, 45141 Essen, Germany

**Keywords:** bryophytes, helophytes, macrophyte communities, hydromorphology, multiple stressors

## Abstract

Upland rivers across Europe still exhibit undisturbed conditions and represent a treasure that we cannot afford to lose. We hypothesize that the combination of pristine and modified conditions could demonstrate biological responses along the stressor gradients. Thus, the response of aquatic macrophyte communities to anthropogenic stressors along upland rivers in Bulgaria was studied. Six stressors were selected out of 36 parameters grouped into hydromorphological, chemical variables and combined drivers (catchment land use). The stressors strongly affected species richness on the basis of biological type (bryophytes vs. vascular plants) and ecomorphological type (hydrophytes vs. helophytes). Hydrological alteration expressed by the change of the river’s base flow and altered riparian habitats has led to a suppression of bryophytes and a dominance of riverbank plant communities. Seventy-five percent of mountain sites were lacking bryophytes, and the vegetation at semi-mountainous sites was dominated by vascular plants. It can be concluded that hydropeaking, organic and inorganic pollution, and discontinuous urban structures caused important modifications in the aquatic macrophyte assemblages. Macrophyte abundance and the biological and ecomorphological type of aquatic macrophytes reflect multi-stressor effects in upland rivers.

## 1. Introduction

“Good ecological status” within the context of the European Water Framework Directive is currently achieved by only 40% of European rivers [1]. Approximately 40% of the highland and mid-altitude rivers in Europe are affected by hydromorphological pressures [2]. There are numerous anthropogenic activities that contribute to the fact that approximately 60% of surface water bodies do not have a good ecological status. Among those with the most significant impact are agriculture, forestry and aquaculture, hydropower generation, and urban settlements [3]. As for the stressors, almost half of the European water bodies are affected by two or more stressors [3]. Nutrients expressed by nitrogen pollution, hydromorphology, and catchment land use were reported as major factors affecting the ecological status during the period 2004–2009 [4]. This trend has continued in recent years. Physical alterations affecting river hydromorphology and diffuse pollution are among the most widespread stressors simultaneously affecting Europe’s waters [5]. The latest research on European rivers reported that almost all river types are more or less affected by riparian land use, hydrological changes, nutrient enrichment, and the input of toxic substances [1].

The protection and restoration of freshwater ecosystems depend on the reliable detection of the stressors’ impact based on the aquatic biota. Most of the monitoring and assessment methods are linked to nutrient pollution, while only a few assess the impact of hydromorphological and multiple stressors [5]. A recent review of papers dedicated to aquatic ecology revealed that nutrient stress was studied in 71% to 98% of multi-stress situations [6]. Despite an increasing number of scientific investigations, there are still open questions regarding pathways between stressors, biota, and ecological status. Another important issue is the assessment of the effects of simultaneous stressors and their possible antagony, additivity, and synergy [7].

Aquatic macrophytes have the ability to give a complex response to multiple stress factors and are recommended for the assessment of multiple stressors. Macrophyte-based metrics were found to be useful for assessing ecological quality at the river basin scale [8]. Macrophytes showed significant correlations with habitat degradation and particularly with woody riparian vegetation [9]. A strong correlation between macrophyte metrics, land-use gradient, and eutrophication/organic pollution in European mountain streams has been documented [10]. Given this background, we collected a database of aquatic macrophytes in Bulgaria (southeastern Europe) and made an attempt to define the driving forces for their communities in upland rivers.

Rivers in mountain regions are under less severe impacts than lowland rivers [2]. Nevertheless, land use in the mountains can alter river habitats, change sediment and nutrient regimes, and pose certain toxic risks. Moreover, a stronger dependence of aquatic communities on the surrounding terrestrial ecosystem is expected in upland rivers [9]. In this study, we look into the stressors affecting mountain rivers and aquatic macrophyte communities as a tool to detect multi-stressor effects in these valuable aquatic ecosystems.

Bulgaria is characterized by its relief diversity and rich geological history. The Stara planina and Rila-Pirin-Rhodopes are the major mountain massifs in Bulgaria [11]. The Balkan Mountains form a phytoclimatic barrier and divide the country into two ecoregions: 12 Pontic Province for northern and 7 Eastern Balkans for southern rivers [12]. Almost all rivers in northern Bulgaria (Ecoregion 12) have their source in the Balkan Mountains [13]. The Peri-Aegean drainage basin (Ecoregion 7) consists of the Struma, Mesta, and Maritsa river systems. The rivers, which flow directly into the Black Sea, belong to the Euxinian (Black Sea) drainage area and are usually short and small. The studied upland rivers belong to four altitude vegetation belts: xerothermic oak forests, xeromesophytic oak and hornbeam forests, beech belt, and coniferous belt [14]. Point source pollution with untreated waste waters and diffuse pollution from agriculture were recognized as the most significant pressures on Bulgarian rivers [15]. However, official monitoring data for Bulgarian rivers show significant gaps.

For this study, we hypothesize that aquatic macrophytes respond to degradation when water bodies are impacted by multiple stressors. Therefore, the aim of this paper was to explore the relationships between aquatic macrophyte community metrics (species richness, abundance, biological and ecomorphological type) in upland rivers in Bulgaria and anthropogenic factors expressed by six stressors: hydromorphological (hydropeaking), chemical variables (BOD_5_, TN, N-NH_4_, TP), and combined drivers (discontinuous urban fabric).

## 2. Results

### 2.1. Studied Sites and Stressors

Thirty-two sites at mountain and semi-mountain rivers in Bulgaria were studied during the vegetation season in 2020 (Figure 1, Annex 1). Sites had mixed carbonate-siliceous bedrock and were located at considerably variable altitudes (between 71 and 1292 m a.s.l.). River water pH was neutral to slightly alkaline.

A summary of the measured stressors (*n* = 36) is given in Annex 2. The most relevant stressors were selected on the basis of their range in the dataset (Figure 2) and co-correlation with other stressors (Spearman correlation coefficient of R_s_ < 0.8).

### 2.2. Aquatic Macrophyte Assemblage Patterns

Fifty-eight macrophyte species were recorded (Table 1); among them were one alga, five mosses, and 52 vascular plants. The most common species were the aquatic moss *Platyhypnidium riparioides* and the helophyte *Mentha longifolia*. Vascular plants had higher species richness than bryophytes (5 species). Among vascular plants, helophytes were the most numerous group (24 species). The number of taxa per river site varied between 1 and 13. Aquatic macrophytes had not been recorded at one site (site 10) due to difficult conditions for sampling during a flood event.

Mountain river sites (national type R03; *n* = 15) were located at high altitude (median: 705 m a.s.l.) and characterized by coarse substrate, rapidly running water, and shaded conditions. Macrophyte communities were dominated by helo- and hygrophytes. Mosses were recorded at only 25% of the sites.

The more variable was the dataset of semi-mountain river sites (national types R04 and R05; *n* = 17). It could be divided into two major subsets using Detrended Correspondence Analysis (DCA). The first subset included the aquatic moss *P. riparioides*; the hydrophytes *Ceratophyllum submersum*, *Lemna minor*, *Myriophyllum spicatum*, *Potamogeton*
*berchtoldii*, and *Potamogeton nodosus*; and the helophytes *Berula erecta*, *Scirpus lacustris*, and *Sparganium erectum*. This assemblage pattern was recorded along small rivers from both northern and southern Bulgaria, at medium altitude (71–493 m a.s.l.), moderate velocity of the current, and diverse dominant substrate (sand, gravel, stones).

The second subset was represented mainly by the helophytes *Echinochloa crus-galli*, *Juncus effusus*, *Lycopus europaeus*, *Lythrum salicaria*, *Mentha aquatica*, *Mentha longifolia*, *Phragmites australis*, *Typha latifolia*, and *Veronica beccabunga* and the hygrophytes *Cyperus longus*, *Epilobium hirsutum*, *Myosoton aquaticum*, *Paspalum paspaloides*, and *Polygonum lapathifolium*. These assemblages were characteristic for small and medium sized rivers in southern Bulgaria at higher altitudes (75–837 m, median 300 m a.s.l.), coarser substrate (stones), and predominantly sunny sites.

Redundancy Analysis indicated a distinct abiotic gradient (Figure 3). The gradient was related to mean river width, flow velocity, channel substrate, and shading, which represent the physical character of a river, distinguishing between more wide and rapid streams (right part of the plot) and shaded and stony streams (left part). Species richness and abundance per site increased in conditions of wider river habitats with rapidly running water, while the number of taxa and their abundance was lower in shaded sites dominated by coarser substrate.

### 2.3. Relationship of Aquatic Macrophyte Communities and Selected Stressors

DCA was used to examine relations between macrophyte communities and stressors. The first and second DCA axes together explained 25% of the species-environmental relation (Figure 4). The first axis had the strongest positive correlation with the percent of discontinuous urban fabric presence, hydropeaking, and total nitrogen. Aquatic macrophyte assemblages in river habitats impacted by man-made structures, hydropeaking, and nitrate loading were characterized by the hydrophytes *Ceratophyllum demersum*, *Myriophyllum spicatum*, *Potamogeton berchtoldii, P. nodosus*, *P. pectinatus,* and *Lemna minor*. Riverbank assemblages at these sites were dominated by *Berula erecta*, *Scirpus lacustris*, and *Sparganium erectum*. The representative taxa of both ecomorphological types were located between the center and the edge of the plot and showed a clear relation with the first axis. The results of the DCA also suggested two homogeneous groups of sites: The larger group (close to the center of the plot) was positioned along gradients of ammonium nitrogen, total phosphorus, and organic matter. These sites featured no hydrophytes. The second group was positioned along the rest of the stressors and included mainly highly modified water bodies. Upland river sites under multiple stress lose their aquatic and rheophilous bryophyte communities. Under nutrient stress, mainly bank assemblages were supported, while sites with pronounced stress of both hydromorphological and chemical disturbance also provided conditions for the development of atypical vascular aquatic plant assemblages.

Multiple regression analysis showed a strong relation of macrophyte abundance (R^2^ = 0.41; *p* < 0.05) with the six stressors. Under conditions of increasing total phosphorus, macrophyte abundance increased, while in river sites rich in total nitrogen the abundance decreased. River sites with the highest nitrogen values supported communities of helophytes and hygrophytes, while floating and submerged aquatic macrophytes were absent or recorded at low abundances. High nitrogen concentrations resulted in a decline of aquatic macrophyte abundance.

## 3. Discussion

### 3.1. Aquatic Macrophyte Response to Multiple Stressors in Bulgarian Upland Rivers

Aquatic macrophytes along upland rivers in Bulgaria respond to multiple stressors through community metrics: species richness, abundance, and biological and ecomorphological type. Hydropeaking; increasing concentrations of nitrogen, phosphorus, and organic matter; and discontinuous urban fabric land use (as a driver of the deteriorated river conditions) correlated with macrophyte abundance. Our results agree with the research of Lemm et al. [1], who reported three stressor categories, ‘hydro-morphology’, ‘nutrients’, and ‘toxic substances’, as affecting the ecological status of European rivers in a ratio of 1.5 to 1.3 to 1.0.

Noteworthy is the fact that the sites dominated by mosses (*n* = 4) had the lowest level of stressors and related drivers; no hydropeaking and discontinuous man-made structures were observed and the median concentrations of chemical variables (nutrients and BOD_5_) were in line with near-natural conditions for upland streams [16].

River sections with bryophyte communities had between 4 and 12 species per site, while the medium total species richness at the studied sites was 8. This supports the common emphasis on macrophyte species composition rather than on species richness to assess ecosystem quality. In addition, undisturbed communities were dominated by stiff rheophilous aquatic mosses forming large carpets (as was previously reported for other river types [17]).

Our results for higher species richness and abundance in predominantly sunny, wider rivers with fine substrate reflected the fact that typical upland bryophyte communities become replaced by assemblages dominated by helo- and hygrophytes, as well as by aquatic macrophyte species characteristic for lowland rivers, such as *Callitriche stagnalis*, *Ceratophyllum demersum*, *Lemna minor*, *Myriophyllum spicatum*, and *Spirodela polyrhiza*. Native vegetation patterns in mountain and upland streams in Central Europe included mostly bryophytes, while vascular submerged plants (e.g., *C. stagnalis*, *C. demersum*) and vascular free-floating plants (*L. minor*, *S. polyrhiza*) occurred in smaller numbers [1]. Similar results on reference mountain rivers in Poland and Slovakia dominated by bryophytes were reported by Szoszkiewicz et al. [18]. Based on the above, the low number of bryophyte taxa (*n* = 5) and their limited distribution (at about 22% of the sites) along the studied rivers can be attributed to human activities, namely hydromorphological modifications and water pollution.

### 3.2. Effects of Individual Stressors (and Related Drivers)

Most of the streams located in the upland region in Bulgaria are influenced by the electricity production of small hydropower plants, causing periodic and rapid flow fluctuations (hydropeaking). Flow alterations and shifts between submergence and drainage most likely suppress aquatic macrophyte reproduction and colonization success. Bryophytes are restricted to stable substrates and, in case of disturbance, a long time for recolonization is needed [19]. Weakly-rooted hydrophytes (e.g., *Myriophyllum spicatum*) are also supposed to be intolerant to hydropeaking. An additional negative factor for macrophytes is the transportation of suspended sediments. The interrupted flow regime can favor only species that are easily dispersed, flexible, flood-tolerant, and amphibious [20], i.e., within this study identified as bank assemblages of *Berula erecta*, *Sparganium erectum*, *Scirpus lacustris*, and *Typha latifolia* at river sites impacted by hydropeaking. Substrate and water level homogenization as a result of hydropower plant operation was previously reported as a key factor leading to replacement of native macrophyte communities of aquatic bryophytes by vascular plant species (*Elodea canadensis*, *Myriophyllum spicatum*, *Potamogeton* spp.), caused by both physical disturbance and elevated nutrient concentrations [21].

The floristic richness in the taxonomic group of bryophytes and specific morphologic groups (particularly helophytes) was reported to highlight differences in channelized streams [17,22]. The observed macrophyte response to habitat alteration was previously reported in investigations demonstrating the response of macrophyte metrics to habitat alterations investigated in European mountain rivers [23].

Higher percentages of discontinuous urban fabric land use (buildings, roads, artificial areas) are known to cause elevated nutrient pollution. Aquatic macrophyte communities respond with an increase in nutrient-tolerant species or species preferring high concentrations of phosphorus and nitrogen (*Potamogeton nodosus*, *P. pectinatus*, *Ceratophyllum demersum*, *Lemna minor*, and *Spirodela polyrhiza*). These hydrophytes have the ability to exploit surface water nitrogen and phosphorus and to form large canopies [24].

Our finding of declining macrophyte abundance with increasing nitrogen concentrations is in line with findings that nutrient enrichment explained a higher share of deviance in mountain rivers than in lowland rivers; Lemm et al. [1] related this outcome to the more pronounced gradients of nutrient enrichment in mountainous catchments in contrast to European lowlands, which are consistently burdened by intensive human land-use stressors.

### 3.3. The role of Macrophyte Assessment and Upland River Management

Our results confirmed that the most widespread stressors simultaneously affecting Europe’s rivers are physical alterations (in particular, hydromorphology) and diffuse pollution [5]. Hydromorphological and chemical stressors, as well as combined drivers, have led to changes in macrophyte communities in upland rivers in Bulgaria. The upland rivers in near-natural conditions were dominated by bryophytes and characterized by relatively low species richness and abundance. Under multiple stressors, aquatic macrophyte assemblages showed a divergent response of the metric species richness. Biological type appeared to be a more reliable response variable, and species composition showed an unambiguous response.

With respect to effective river management, macrophyte indices based on indicator species (e.g., taxonomic composition and abundance) offer adequate tools for the assessment of ecological river status. However, strong relations with multiple stressors still have to be proven for the diagnosed indicator species.

Our observations highlight two further requirements: First, monitoring and assessment schemes should be based on environment-biology relationships considering multiple stressors and possible interactive effects. For instance, hydromorphological stress could alter the sensitivity to nutrient stress [6]. Second, aquatic communities under undisturbed conditions should be documented in order to better understand the mechanisms of stressor effects when compared against undisturbed conditions.

## 4. Materials and Methods

### 4.1. Study Area and River Sites

Aquatic macrophytes were surveyed at 32 upland river sites in Bulgaria during 2020 (Figure 1, Annex 1). The survey covered 15 sites along mountain and 12 along semi-mountain rivers in Ecoregion 7, eastern Balkans, and 5 sites at semi-mountain in Ecoregion 12, Pontic province. The sampling sites were selected from the national monitoring network and are representative of the types and the variation in human pressures (approximately one-third of the sites were designated as highly modified water bodies), and they include a number of potentially undisturbed sites.

### 4.2. Field Survey and Additional Data Acquisition

Plant species were recorded along a 100 m long section following a zigzag pattern. The nomenclature followed Hill et al. [25] for mosses and Euro + Med [26] for vascular plants. The species abundance was recorded using a five-level scale [27]: 1 = very rare, 2 = infrequent, 3 = common, 4 = frequent, 5 = abundant, predominant.

Additionally, four abiotic parameters were recorded: flow velocity, shading, substrate type, and mean depth. They were determined in a semi-quantitative way using class scales, to enable a fast and easy field application, as used by Schaumburg et al. [28,29]. Shading was noted based on a 5-degree scale (1 = completely sunny, 2 = sunny, 3 = partly overcast, 4 = half shaded, 5 = completely shaded). Velocity of flow was recorded via a six-point scale: 1 = not visible, 2 = barely visible, 3 = slowly running, 4 = rapidly running (current with moderate turbulences), 5 = rapidly running (turbulently running), 6 = torrential. The substratum conditions at the sampling site were classified in 5% steps according to an eight-point scale: % mud, % clay/loam (<0.063 mm), % sand (0.063–2.0 mm), % fine/medium gravel (2.0–6.3/6.3–20 mm), % coarse gravel (20–63 mm), % stones (63–200 mm), % boulders (>200 mm), and % organic/peat. Mean depth was noted via a three-degree scale (I = 0–30 cm, II = 30–100 cm, III > 100 cm).

In situ measurements of acidity (pH) and electrical conductivity (C, µS cm^−1^) of river water were taken using a calibrated WTW pH/Conductivity meter. Biochemical oxygen demand, nitrogen, and phosphorus compounds were analyzed following adopted standards in an accredited laboratory (Aquaterratest Lab, Sofia, Bulgaria).

### 4.3. Stressors

The term stressor was applied following Perujo et al. [30]. Multiple stressors included (i) hydromorphological (No1–10, Annex 2), (ii) chemical variables (No11–18, Annex 2), and (iii) combined drivers (No19–36, Annex 2).

Data for the first group of the studied anthropogenic stressors were collected during the sampling campaign in 2020. This group cause hydrological and/or flow modifications due to water abstraction and impoundment and hydropeaking, as well as morphological impairment due to damming, channelization, alterations in riparian vegetation, habitat and landscape. These stressors were classified based on the collected information, in relation to their degree of alteration from the natural state via a three-category assessment scheme following the European intercalibration approach [17].

The second group included chemical stressors affecting river biota, such as nutrients.

The third group, known as combined drivers, incorporates 18 categories of land use calculated on a catchment scale using Corine Land Cover (CLC) data [31].

### 4.4. Analysis of Aquatic Macrophytes, Stressors and Their Relation

Two-Way Indicator Species Analysis, TWINSPAN [32], Canoco [33], and STATISTICA software were applied to the datasets.

To classify aquatic macrophyte data from the variable semi-mountain rivers communities, TWINSPAN was used. Plant data were standardized to a presence/absence scale.

The impacts of environmental variables on aquatic macrophyte communities were examined by constrained ordination. The statistical significance of the relationships between biological dataset and stressor variables was evaluated using the Monte Carlo permutation test (499 permutations). Redundancy analysis (RDA) was performed to reveal the variation in the basic community’s metrics along physical river characteristics (mean width, flow velocity, substrate, and shading). Detrended correspondence analysis (DCA) was applied to examine the impact of selected stressors and drivers: hydropeaking, BOD_5_, nitrogen and its inorganic forms, phosphorus, and discontinuous urban fabric. Rare species (having only one occurrence) were excluded.

To establish the correlations, multiple regression analysis of basic community metrics and hydromorphological, chemical variables, and combined stressors drivers (catchment land use) was performed. A coefficient of determination (R^2^), Pearson correlation coefficient (P), and Spearman’s Correlation Coefficient (R_s_) were calculated to estimate variance explanation and to test for significance.

## 5. Conclusions

The drivers represented by urban structures on significant surfaces in a discontinuous spatial pattern and the anthropogenic stressors hydropeaking, organic pollution, and the loads of nitrogen and its inorganic forms appeared to have a considerable effect on rivers’ aquatic macrophyte communities. Their impact was revealed on upland rivers in Bulgaria. Vascular plants were the richest taxonomic group, and helophytes were the richest morphologic group. Three assemblage patterns were distinguished: (i) high altitude mountain river communities dominated by helo- and hygrophytes; (ii) communities of aquatic mosses, submerged and free-floating vascular plants, and helophytes at medium altitude; (iii) assemblages dominated entirely by helophytes and hygrophytes along upland rivers.

Higher species richness and abundance was recorded at predominantly sunny habitats with fine substrate. A clear trend of decreasing floristic abundance was assessed along the gradient of nitrogen pollution. Macrophyte abundance was positively correlated with increased phosphorus concentrations. Under anthropogenic stressors such as urban activities and loads of nitrogen and phosphorus, macrophyte assemblages included the hydrophytes *Potamogeton nodosus*, *P. pectinatus*, *Ceratophyllum demersum*, *Myriophyllum spicatum*, and *Lemna minor*. Emergent macrophyte species (*Berula erecta*, *Scirpus lacustris*, *Sparganium erectum*) dominated at river sites under hydropeaking impact. The achieved results illustrated that less than one-third of the studied river sites were relatively unaffected by stressors. Although merely a case study, we found form our results, a signal that the anthropogenic pressure on upland rivers is growing, and the places where unaffected conditions can be registered are rapidly decreasing. Aquatic macrophyte-based markers can be used to detect, measure, and track changes in the upland rivers in southeastern Europe.

## Figures and Tables

**Figure 1 plants-10-02708-f001:**
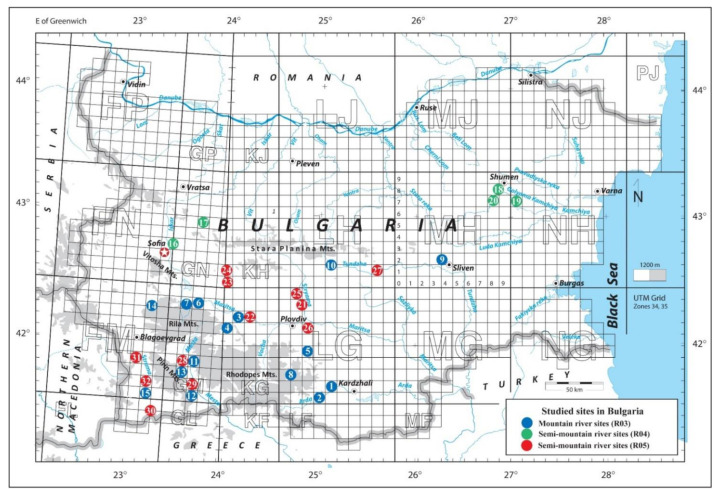
Location of the studied river sites. Refer to Annex 1 for the numbering and national types of the sites.

**Figure 2 plants-10-02708-f002:**
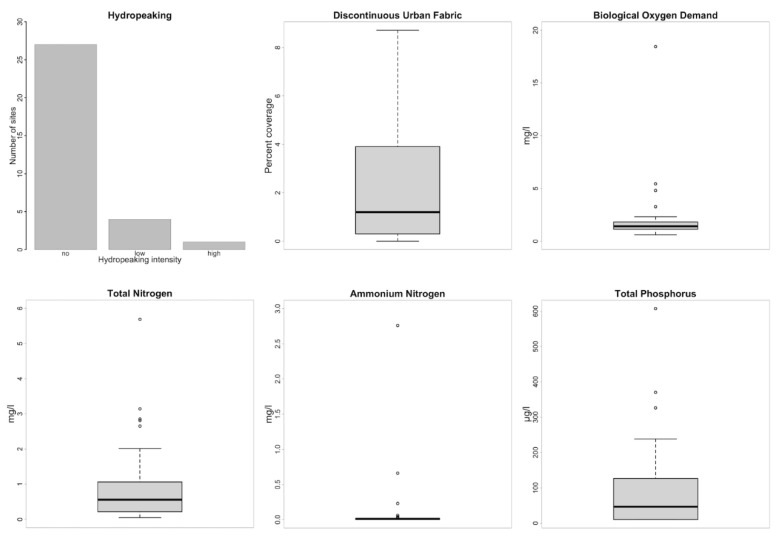
Frequency and distribution of the six anthropogenic stressors analyzed in this study.

**Figure 3 plants-10-02708-f003:**
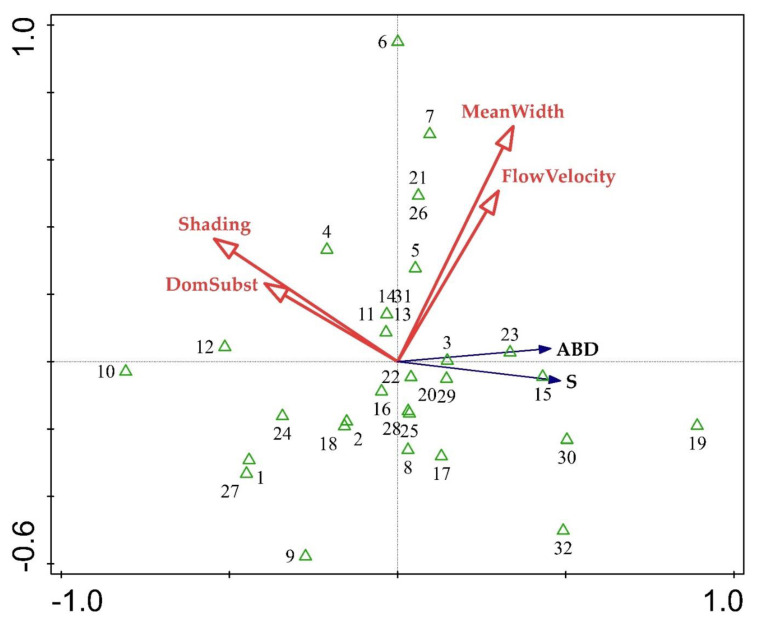
RDA ordination tri-plot with species richness (S), macrophyte abundance (ABD), and four abiotic characteristics. Refer to Annex 1 for the numbering of the sites.

**Figure 4 plants-10-02708-f004:**
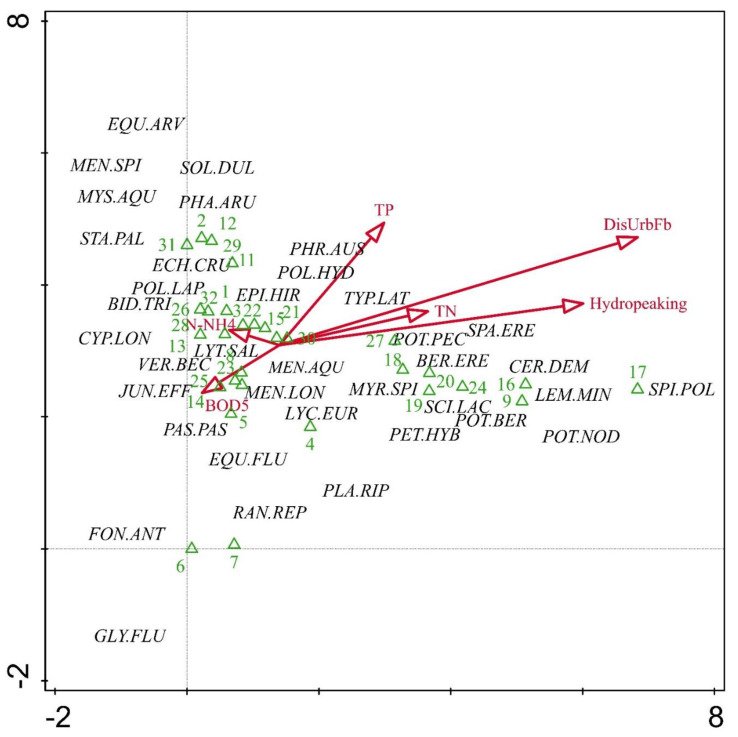
DCA ordination plot with selected environmental factors (DisUrbFb: discontinuous urban fabric %, TN: total nitrogen, TP: total phosphorus, N-NH^4^: ammonium nitrogen, BOD_5_: biological oxygen demand). Eigenvalue of the first axis: 0.747; second axis: 0.563. Refer to Table 1 for species codes.

**Table 1 plants-10-02708-t001:** List of species and their codes.

Algae	Code
*Lemanea fluviatilis*	LEM.FLU
**Mosses**	
*Brachythecium rivulare* Schimp.	BRA.RIV
*Cratoneuron filicinum* (Hedw.) Spruce	CRA.FIL
*Fontinalis antipyretica* Hedw.	FON.ANT
*Leptodictyum riparium* (Hedw.) Warnst.	LEP.RIP
*Platyhypnidium riparioides* (Hedw.) Dixon	PLA.RIP
**Vascular plants**	
**Pteridophytes**	
*Equisetum arvense* L.	EQU.ARV
*Equisetum fluviatile* L.	EQU.FLU
*Equisetum sylvaticum* L.	EQU.SYL
*Equisetum telmateia* Ehrh.	EQU.TEL
**Hydrophytes**	
*Callitriche stagnalis* Scop.	CAL.STA
*Ceratophyllum demersum* L.	CER.DEM
*Ceratophyllum submersum* L.	CER.SUB
*Elodea canadensis* Michx.	ELO.CAN
*Lemna minor* L.	LEM.MIN
*Myriophyllum spicatum* L.	MYR.SPI
*Potamogeton berchtoldii* Fieber	POT.BER
*Potamogeton crispus* L.	POT.CRI
*Potamogeton nodosus* Poir.	POT.NOD
*Potamogeton pectinatus* L.	POT.PEC
*Potamogeton perfoliatus* L.	POT.PER
*Potamogeton pusillus* L.	POT.PUS
*Ranunculus trichophyllus* Chaix	RAN.TRI
*Spirodela polyrhiza* (L.) Schleid.	SPI.POL
**Hygrophytes**	
*Bidens tripartitus* L.	BID.TRI
*Cyperus longus* L.	CYP.LON
*Epilobium hirsutum* L.	EPI.HIR
*Myosoton aquaticum* (L.) Moench	MYO.AQU
*Paspalum paspalodes* (Michx.) Scribn.	PAS.PAS
*Petasites hybridus* (L.) G. Gaertn. & al.	PET.HYB
*Polygonum lapathifolium* L.	POL.LAP
*Polygonum mite* Schrank	POL.MIT
*Ranunculus repens* L.	RAN.REP
*Solanum dulcamara* L.	SOL.DUL
**Helophytes**	
*Alisma lanceolatum* With.	ALI.LAN
*Alisma plantago-aquatica* L.	ALI.PLA
*Berula erecta* (Huds.) Coville	BER.ERE
*Cyperus fuscus* L.	CYP.FUS
*Echinochloa crus-galli* (L.) P. Beauv.	ECH.CRU
*Glyceria fluitans* (L.) R. Br.	GLY.FLU
*Juncus effusus* L.	JUN.EFF
*Lycopus europaeus* L.	LYC.EUR
*Lythrum salicaria* L.	LYT.SAL
*Mentha aquatica* L.	MEN.AQU
*Mentha longifolia* (L.) L.	MEN.LON
*Mentha spicata* L.	MEN.SPI
*Nasturtium officinale* W. T. Aiton	NAS.OFF
*Phalaris arundinacea* L.	PHA.ARU
*Phragmites australis* L.	PHR.AUS
*Polygonum hydropiper* L.	POL.HYD
*Rorippa amphibia* (L.) Besser	ROR.AMP
*Sagittaria latifolia* Willd.	SAG.LAT
*Scirpus lacustris* L.	SCI.LAC
*Sparganium erectum* L.	SPA.ERE
*Stachys palustris* L.	STA.PAL
*Typha angustifolia* L.	TYP.ANG
*Typha latifolia* L.	TYP.LAT
*Veronica beccabunga* L.	VER.BEC

## Data Availability

The data presented in this study are available in Annex 1 and 2.

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
