# Peer review of "Anthropogenic Stressors in Upland Rivers: Aquatic Macrophyte Responses. A Case Study from Bulgaria"

_plants, 2021, doi:10.3390/plants10122708_

Round 1
Reviewer 1 Report
The manuscript brings interesting novelties in the issue of aquatic macrophyte responses to antrophogenic stressors. It hihglinghts interesting finding and confirm the significance of monitoring macrophyte community changes in view of waterbody changes. Moreover, it is very important having in mind that very few such a studies had been carried out in the regions of South-Eastren Europe (i.e. the Balkans). Additionally, such data are missing not only from the point of conservation but also from the point of managing mountain waters. The quality of the graphics are rather low and especially figur 2 (the irst one) is hard to follow, so please give the one in higher resolution and larger if possible. Please pay attention that You have twice Fig. 2 within the manuscript. Please, change. In general, the manuscript is interesting for wider readership and acceptable after minor changes.
Author Response
First of all, we would like to thank the Reviewer 1 for the for the positive evaluation and precise comments. All Reviewer’s recommendations were reflected. (Reviewer’s comments are shown in Italic).
The manuscript brings interesting novelties in the issue of aquatic macrophyte responses to antrophogenic stressors. It hihglinghts interesting finding and confirm the significance of monitoring macrophyte community changes in view of waterbody changes. Moreover, it is very important having in mind that very few such a studies had been carried out in the regions of South-Eastren Europe (i.e. the Balkans). Additionally, such data are missing not only from the point of conservation but also from the point of managing mountain waters.
The quality of the graphics are rather low and especially figur 2 (the irst one) is hard to follow, so please give the one in higher resolution and larger if possible.
Thanks to your recommendation Figure 2 was presented in a higher resolution (1000 dpi). The file size is 8.25 MB and if the Journal decides, during the pre-press preparation, it can be presented in a larger format. Figures 1, 3 and 4 were also improved.
Please pay attention that You have twice Fig. 2 within the manuscript. Please, change.
Thank you, we have adjusted the numbering of the figures.
The Cover Letter is attached.

Reviewer 2 Report
The authors greatly improved the manuscript. However, it is still necessary to add a map of the research location against the background of the map of Europe and Bulgaria. In its present form, Fig. 1 is completely illegible. Description 2.1. Studied sites and stressors should be included in the materials and methods section, not in the results. Descriptions contained in "materials and methods" should be divided into sub-sections, because in their present form they are difficult to read. This chapter requires a thorough redrafting.
Author Response
We thank the Reviewer 2 for the positive evaluation of the manuscript and valuable suggestions. Below you can find the answers to your comments. (Reviewer’s comments are shown in Italic).
The authors greatly improved the manuscript. However, it is still necessary to add a map of the research location against the background of the map of Europe and Bulgaria. In its present form, Fig. 1 is completely illegible.
Thank you – a new map was provided.
Description 2.1. Studied sites and stressors should be included in the materials and methods section, not in the results.
Thank you - Although we believe we understand your consideration, we leave this subsection in the Results-part because (i) the studied sites are a result of pre-selection, as pointed out in the Material and methods section and (ii) the subsection contains information resulting from the study: Annex 1 contains measured pH and temperature of the river water measured in-situ; Annex 2 contains the range of the selected environmental variables (physical stressors, physical and water chemical parameters, combined drivers) obtained during the study.
Descriptions contained in "materials and methods" should be divided into sub-sections, because in their present form they are difficult to read. This chapter requires a thorough redrafting.
Thank you for the valuable comment – four subsections were added and the chapter was revised.
The Cover Letter is attached.

This manuscript is a resubmission of an earlier submission. The following is a list of the peer review reports and author responses from that submission.
Round 1
Reviewer 1 Report
The manuscript of Gana Gecheva and colleagues describes a study undertaken at Bulgarian mountain rivers, linking macrophyte community composition to anthropogenic pressures. In its current form, I cannot recommend the manuscript for publication. It very much reads like a project-report that explains data collection and analysis, yet fails to establish a broader relevance of the findings for an interested readership. It mainly lacks a compelling storyline/narrative that readers can identify with.
In the following, I provide some suggestions that may help to improve the manuscript.
1. One possible storyline to be followed could be:
(i) Mountain rivers are affected by multiple stressors (featuring typical stressor combinations, e.g. stemming from hydropower or urbanization).
(ii) Macrophyte communities allow for detecting these multi-stressor effects.
(iii) The Bulgarian case-study shows, how.
Once you selected and “sharpened” your storyline, you need to re-design the entire manuscript around this storyline.
2. The study is solely based on Bulgarian data; thus, the authors should carefully describe the specific settings: What are the natural features (e.g. climatic, geologic, pedologic) of the Bulgarian mountains? Do you cover different types of Bulgarian mountains with regard to the characteristics of their natural environment? What made you select the monitoring sites, and why do you think they are representative of mountain rivers? How do the studied river types look like (perhaps provide pictures)? Where are they located (on the map)? What are the main types of anthropogenic activities? How do these activities affect the rivers (and monitoring stations)? What are the main abiotic effects of these anthropogenic activities on the monitoring sites?
This would make the Bulgarian case-study much stronger. Furthermore, consider providing the data in an accessible online-repository.
3. You need to strengthen your discussion. Instead of “dragging in” seemingly related analyses showing correlations of ecological status with anthropogenic stressors (e.g. Lemm et al. 2021), focus on your storyline:
- What do we know about multiple stressors in mountain streams (relevant to macrophytes), compared to lowland streams (e.g. different drivers)?
- What do we know about the potential of macrophytes to indicate multiple stressors? How do they do this indication? Does such an indication make sense (including: Do macrophyte at all effectively indicate stressors, given that the species richness is often pretty low)?
- Does it help in river management? Do we need other indicative techniques (e.g. diagnostic methods), and do macrophytes offer support here (see, for instance, Lemm et al. 2019 STOTEN 651: 1105–1113)?
- How do both aspects (mountain river multi-stressor environment & macrophytes bioindication) come together (e.g. compared to lowland river environments)?
4. The analyses you made are based on associations of biological data with putative stressor data. This isn’t an issue per se but needs careful consideration: Ordination plots allow for some kind of “open interpretation”. At least in the discussion section, I would like to see causal interpretations regarding the mechanisms, for instance, why the community reacts to hydropeaking this way. This also needs to include an explanation of how “catchment land use” is affecting the macrophyte community. I consider “land use” a driver exerting certain pressures on the aquatic environment that manifest in deteriorated state variables, which affect the biology. Please be more critical with your interpretation given the available data. Furthermore, where are the TWINSPAN results?
5. Some minor comments:
- Check the use of terms: You seem to use “mountain”, “semi-mountain”, “upland” interchangeably (or it’s at least not clear to me why you use different terms here – re-consider if this is really necessary … perhaps you need to get away from the formal BG-river-typology-thinking)
- Check the use of terms: You seem to use “pressure” and “stressor” interchangeably.
I recently wrote: “The choice of the terms ‘pressure’ and ‘stressor’ is not arbitrary; rather, it is rooted in their different uses in environmental management and ecological science. The DPSIR (i.e., Driver- Pressure-State-Impact-Response) adaptive management framework (EEA, 1999) defines Pressures as the direct effects of a Driver; that is, an anthropogenic activity (e.g., agriculture, industry) or climate change phenomenon (e.g., climate warming, changes in precipitation). Pressures affect the ecosystem’s State; that is, its physical, chemical, and biological characteristics. State changes may result in effects on ecosystem characteristics valued by man (Impact), triggering social Response to mitigate these effects.
The term ‘stressor’ is not used in the DPSIR framework, and this circumstance often promotes confusion among managers and scientists. One ultimate aim of scientific research is to infer causality by detecting the mechanistic pathways in which the stressors affect the receptors. From the scientist’s viewpoint, stressors are thus (putative) causes in a cause- and-effect chain. This places stressors within the Pressure or State element of the DPSIR frame- work, depending on which causal parameters are investigated.”
You may also check Perujo et al. (2021) STOTEN 777: 146112.
- Very few language problems show up in your manuscript, mainly related to missing definitive articles (“the”) and slightly odd sentence constructions here and there. Perhaps you can check the manuscript by a native English speaker before submission?
- The numbers in each separate row of a table should have the same number of decimal digits. And check for ‘points’ as decimal separators, not ‘commas’. Furthermore, always provide full information on variables and their abbreviations in the table/figure caption.
- “Estuary” is where the river meets the sea (= check the naming of your sampling sites).
- Define each of your classes in your 3-point pressure-scale.
- I warmly recommend the following paper for helping to improve your writing style: Plaxco, K.W., 2010. The art of writing science. Protein Sci. 19, 2261–2266. https://doi.org/10.1002/pro.514
Author Response
To the Reviewer 1
Thank you very much for the constructive review. Your concerns were addressed and in the attached Cover Letter you can find our answers to your comments.

Reviewer 2 Report
The manuscript presented for review concerns the issue of anthropogenic pressure in mountain and semi-mountain rivers in relation to aquatic macrophyte responses. It is an important topic due to the risk of anthroporesia on the aquatic environment, as well as the issues of water quality. Both issues are very topical and important. However, the mauscript itself disappointed me. It is hard to read, does not contain information contained in the title, and is devoid of the necessary graphic documentation. Below are some detailed notes on the manuscript: 1. Introduction - there is no information on human pressure on rivers - the authors treated this issue more than generally 2.line 126? Table 1 for number of sites. 3. The manuscript should be supplemented with maps / photos 4. In my opinion, the anthropogenic pressure should be described first - the research results, and then the species 5.in point 2.1 there is no information about the pressure (which is in the title) 6. The materials and methods need to be detailed, as in their current form they cannot be repeated 7. The topic of anthroporesis is treated very generally, there is a lack of descriptions, information, etc. The authors focus mainly on species, which means that the topic given in the title of the manuscript is not exhausted.
Author Response
We sincerely thank the Reviewer 2 for the evaluation of the manuscript and valuable comments. We appreciate the inputs given. In the attached Cover Letter You can find our answers to your specific comments.

Round 2
Reviewer 1 Report
Thanks for providing a revised manuscript-version. I went through the paper very thoroughly and have the following suggestions for major revisions (as I cannot accept the current version "ready for publication").- The assumption stated at the beginning of the abstract and returned to in the main text (upland rivers considered being mostly pristine) is in conflict with existing evidence. A good overview on the status of "upland rivers" is provided in Lyche Solheim et al. (2019) STOTEN 697: 134043. Figure 6 distinguishes between different mid-altitude and highland river types, demonstrating considerable pressure levels and impaired ecological status. I don't think that the "pristine story" is the right narrative here - you probably better focus on the differences between lowlands and uplands: The latter still features undisturbed conditions alongside highly polluted/modified conditions (featuring stressor gradients that allow for investigating into biological responses). The drivers differ between lowlands and uplands: The latter being impacted by hydropower (and not so much agriculture) and (untreated) point source pollution.
- Please remove the first sentence of the introduction, as the reference to ecosystem services is not relevant for this paragraph/paper.
- I try to summarise the sequence of your story in the introduction: (i) ecological status of rivers in Europe and pressures causing the status; (ii) multiple stressors causing ecological status deterioration; (iii) macrophytes are suitable bioindicators under multi-stressor conditions; (iv) condition of upland rivers; (v) Bulgarian upland rivers' environmental setting and pressures; (vi) hypothesis: macrophytes indicate multi-stressors in upland rivers. Regarding (i): Please use the facts of EEA (2018) at https://www.eea.europa.eu/publications/state-of-water. Please talk about "pressures" when using WFD terminology and facts. Regarding (iii): You should re-write this paragraph into a convincing statement. Details such as significance levels should be avoided in the introduction. The reference to Kaijser et al. (2021) is misplaced here. Rather concentrate on convincing the reader that macrophytes are good indicators of multiple anthropogenic stressors (including a statement about what makes them better than / different from invertebrates or fishes). Regarding (iv): As written above, your assumption is not backed by the existing evidence. Please re-write accordingly. Regarding (v): Please concentrate only on the altitude zone that you have investigated. You should address the distinction of the national river typology (but try to explain the reader the environmental differences of these types, not just stating their names). You don't need to state the numbers of the ecoregions. Do you need to provide all the different names of the rivers (like "Maritsa/Evros/Merich") or can you agree on one common name (without insulting your neighbours too much)? Please provide the pressures acting on the Bulgarian rivers located in the region/altitude zone that you have investigated. Don't provide a general overview for entire Bulgaria, and don't state "unknown anthropogenic pressures" as this is not really informative (and somehow too much WFD-language). Regarding (vi): The statement "indirect human impact on upland rivers is spread on a much broader scale" is very vague, also given that the evidence of human impacts on upland systems seems pretty obvious. The statement "provide reliable diagnosing of degradation" should be re-formulated, as a diagnosis in the strict sense (detecting causes from "symptoms") has not been done. You have rather "associated" patterns in the macrophyte community to environmental stressors (which is a valid approach) -- you should perhaps talk about "demonstrating the response of the macrophyte community to ...". To avoid the terminological issue of "pressures", "drivers" and "stressors", you could generally refer to "anthropogenic factors including ...". Later in the manuscript, you often refer to "basic community metrics". I suggest that you introduce this term already in the introduction.
- The section on "2.1. Studied sites and stressors " does not belong into the results-part but the methods-part.
- Figure 1: This map needs to be re-drawn with a focus to increase the contrasts -- I suggest to use a black-and-white map-type with grey shades to indicate the altitude level, dots and names for the major cities and black lines resembling the major rivers.
- The caption of Table 1 needs to include the names of the national river types. The column header of the coordinates needs to state "Latitude" and "Longitude".
- I suggest that you check the environmental variables for their distribution in the dataset and perform a correlation analysis between the variables (preferably Spearman), then exclude those variables that (i) do not show a homogeneous distribution across the gradient and (ii) are highly correlated with another variable. This way, you "clean up" your dataset and provide a more sound understanding of the actual pressure gradients driving your community. Please be inspired by Feld et al. (2016) STOTEN 573: 1320–1339 in doing so.
- Lines 119f: Write "by shaded conditions" instead of "mostly in the shade".
- Lines 122f: It is unclear what you mean by "It was divided at the third level of the classification analyses into 3 subsets." This point actually raises three further comments: (1) Reading through this paragraph makes me think: You applied Correspondence Analysis. This is not introduced in the methods-section (see below for my further comments regarding the methods-section). (2) From Figure 2 it seems that you treated all sites together ... why do you distinguish between national types in the text of your results? You should avoid introducing some "a priori" type-classification -- this allows you to work with the entire dataset and this provides insights into the usefulness of the national type definitions for macrophytes. (3) Referring to results based on single-site evidence ("at a small river in South Bulgaria (site 17)") is to be avoided. You should rather explain your results focussing on the main community-differences (i.e. being represented by more than a single site). The single-site "outlier" can be mentioned later in the description. Furthermore, there is generally no need to name all relevant sites in the results-section.
- Lines 156f: This first sentence belongs to the methods-section. Please state the results here and not repeat the methods. Furthermore, don't use "first group of variables" but give a name to this group so that the reader doesn't need to refer back to the table. Yet, you may have to exclude variables anyway after you did the correlation analysis mentioned above.
- Lines 157f: Now you refer to "multiple regression analysis" which is also not specified in the methods-section (see below for my further comments regarding the methods-section).
- General remark: Write either "phosphorus" or "phosphorous" throughout the manuscript.
- Line 159: Do you refer to BOD when stating "organic matter"? Please clarify.
- Lines 159ff: Did you perform three separate multiple regression analyses? If yes, why did you do so? I would not run separate analyses to the same dataset including different stressor variables. Please refer to Feld et al. (2016) STOTEN 573: 1320–1339 for a good methodological guidance.
- Lines 168ff: The first two sentences belong to the methods-section. I would advise against stating the eigenvalues so prominently: They should be mentioned in the caption of Figure 3, but in the main text you should only refer to the percentage of explanation. Write "was positively related to" instead of "was positively contributed by".
- Lines 174ff: The sentence starting "Aquatic macrophyte assemblages ..." is far too long -- please shorten.
- Please use the preposition "at" when referring to a site (e.g. "at the site" not "in the site").
- Some further remarks on the discussion-section: 1. You need an introductory paragraph directly related to your hypothesis stated in the introduction. The reader likes to know: Did they found what they were expecting? And according to your hypothesis (which needs to be re-framed, as commented above), the impact of multiple human stressors on the macrophytes was scrutinized. Against this background, you can address the community characteristics found in the second paragraph of your discussion. 2. Lines 201f: I would avoid providing too detailed results-presentations in the discussion ... here you can perhaps simply state that chemical concentrations of nutrients and BOD are in line with near-natural conditions for upland streams (referring to Pardo et al. 2012 STOTEN 420: 33–42). 3. Lines 203f: Suddenly, a "jump" in the text-logic occurs when referring to "what we need to understand better". Please be careful with the storyline in your paper: One paragraph, one topic (see Plaxco 2010). The reader will hardly follow "jumps" in the text ... I would suggest that you organise your discussion by sub-headers, which help the reader to orient. Your focus can be: (i) multi-stressor macrophyte effects in BG upland rivers, (ii) effects of individual stressors (and related drivers), (iii) implications for/role of macrophyte assessment and upland river management -- but these are merely suggestions which require a bit more reflection by you anyway. 4. Lines 205ff: It is unclear what your sentence is meaning. This comment also underlines that you need to re-frame your entire storyline in the discussion (including sub-headers to structure the different topics addressed). 5. Lines 209ff: These sentences are not understandable. You should carefully introduce one idea at a time ... please avoid brining too many different concepts into a single paragraph. 6. Lines 227f: "This also confirmed our hypothesis that human impact on upland rivers has been already spread on a much broader scale than expected." This sentence needs to be re-formulated according to the revision of the hypothesis. 7. Lines 229ff: The three points are rather conclusions than discussion-points. However, I question how these points are actually related to the particular work that you present in the paper. They seem very general and cover very different aspects ... it is hard to understand how you can conclude on such a broad array of recommendations at this point in the discussion ... again another indicator for the need to re-structure the discussion entirely. 8. The following paragraph on hydropeaking "jumps back" into a discussion on the effects after you made the somehow conclusive points above. Please shorten the paragraph (and consider a paragraph-structure described in Plaxco 2010), then re-locate the paragraph within the revised discussion-section. 9. Lines 262ff: This sentence is hardly understandable.
- Lines 309ff: Please do not refer to the names of the software used but provide a step-wise account of the different analyses you performed. Obviously, you did "constraint ordination" (Why did you use both Redundancy Analysis and Canonical Correspondence Analysis?), then you did classification (How did you do this?), then you applied a TWINSPAN-analysis to the classes (Where are the details here? What for?). From reading your results-section, at least two additional analyses were applied: Correspondence analysis and multiple regression analysis. Make sure to provide all necessary details for these analyses as well. You also seem to have separated the dataset of the national river types ... where did you explain this? Why did you do this (e.g. based on which evidence)?
- Consider publishing your basic data in an accessible online-repository such as Github or Zenodo.
- I understand that you consider the "conclusions" being a summary of your findings. I would partly agree that this is beneficial for your text. However, "conclusions" need to reach further ... so please don't simply summarise what you have found but provide the "bottom line" statement related to your work -- some "take home message" that the reader can keep in mind.
Author Response
First of all, we would like to thank the Reviewer 1 for the valuable and precise comments. We consider it important to emphasize that supporting the comments with concrete examples, references and suggestions had greatly helped the revision process. We hope that our revisions reflected the Reviewer’s requirements. Reply point by point in the attachmentThe point-to-point reply is in the attachment (plants-1294855-revised II-Cover Letter.pdf)。

Reviewer 2 Report
The authors mostly took into account the comments of the reviewers. However, I feel unsatisfied with the detailed description of anthroporesia in general, and in this particular case. The descriptions provided are too laconic and not supported by data. Especially that the title includes "anthropopression", so I expect more detailed information and discussions, not just a list of species.
Author Response
We thank the Reviewer 2 for the evaluation of the manuscript.
Following your comment and recommendations of Reviewer 1 we completely revised Results- and Discussion-section.

Round 3
Reviewer 2 Report
The authors greatly improved the manuscript. However, it is still necessary to add a map of the research location against the background of the map of Europe and Bulgaria. In its present form, Fig. 1 is completely illegible. Description 2.1. Studied sites and stressors should be included in the materials and methods section, not in the results. Descriptions contained in "materials and methods" should be divided into sub-sections, because in their present form they are difficult to read. This chapter requires a thorough redrafting.